# Usability of Memes and Humorous Resources in Virtual Learning Environments

Álvaro Antón-Sancho [1], María Nieto-Sobrino [2], Pablo Fernández-Arias [3] and Diego Vergara-Rodríguez [3,*]

1   Department of Mathematics and Experimental Science, Catholic University of Ávila, 05005 Ávila, Spain; alvaro.anton@ucavila.es
2   Department of Education, Catholic University of Ávila, 05005 Ávila, Spain; maria.nieto@ucavila.es
3   Department of Mechanical Engineering, Catholic University of Ávila, 05005 Ávila, Spain; pablo.fernandezarias@ucavila.es
*   Correspondence: diego.vergara@ucavila.es

**Abstract:** This research consists of a quantitative analysis of the perspective of a group of university professors from different areas of knowledge and from 19 different Latin American countries on the use of humor and memes in virtual learning environments (VLEs) in higher education. The data have been obtained from an own-design survey, and the answers have been analyzed in a descriptive and inferential way with the aim of knowing the opinion of the 401 participants (professors) about the didactic effectiveness of humor and the benefits and employability of memes in virtual classrooms. The analysis differentiates the sample by the professors' area of knowledge as the main variable, and by gender, age and teaching experience. As results, the participants give a high evaluation of humorous didactic resources, particularly memes, although the evaluation of their usability in the classroom is intermediate. In this sense, it is shown that the area of knowledge has a significant influence on opinions in this regard.

**Keywords:** humorous resources; meme; virtual learning environment; higher education; quantitative analysis

## 1. Introduction

In recent years, there has been a significant increase in the use of Virtual Learning Environments (VLE) in higher education, which has generated adaptation needs in universities and in continuing teacher training. This strong irruption of VLEs is motivated by several reasons, among which technological development and the needs derived from the COVID-19 crisis stand out [1]. In general, university students are satisfied with the use of this type of environment, although they recognize that there are important limitations. Among them, the needs they pose at a technical level and a certain tendency for the classes carried out through VLEs to be monotonous and boring stand out [2,3].

As a consequence of the above, there is an abundance of research that seeks to develop didactic and methodological resources to achieve an adequate academic and formative performance among university students through the VLE [4]. In this sense, the specialized literature recognizes that it is essential to design mechanisms to capture the attention and facilitate the motivation of students when the teaching–learning process is carried out through these environments [5,6].

The pedagogical need to increase student motivation in the VLE has led to a growing effort to learn about student interests in order to orient these learning environments toward them [7]. This search for motivational elements in virtual teaching that connect with students has led to the use of certain innovative didactic resources, including, for example, humorous resources that have been credited in the literature with improving academic results, as well as increasing interest and participation in the subjects [8,9].

In this sense, learning in a playful way is a motivational incentive for students [10]. This fact finds a solid foundation from the perspective of neuroeducation, which, from the study of the brain and neuroscience, allows for the discovery of new methodologies and didactic resources such as humorous resources that are favorable for teaching in the VLE [11]. This doctrine provides knowledge of the benefits of implementing humorous resources in the educational environment, as well as the neurophysiological foundations involved in the teaching–learning processes [11,12] in order to promote more meaningful learning among students [13]. In this field, neuroeducation deals with the study of brain functioning and the processes of attention, memory, executive functions and emotional development [14–16], which could be improved by using humorous resources in the learning process.

The connection with students is linked to the act of socializing, which generates in humans, as social beings, a sense of pleasure. Extrapolating this to a virtual scenario, the use of visual resources, e.g., emoticons or certain humorous resources, is a novel source of social relations, which is preferred by most of the young population. In this sense, the use of virtual resources is favorable for training, from the point of view of social learning theory [17], and has proven to be especially effective when introduced in virtual learning environments [18]. In this way, humorous resources are interesting tools to favor interaction between professors and students, increasing their motivation and reforming their involvement in the acquisition of knowledge [19–21].

Delving further into aspects of neuroscience related to happiness and pleasure, it is worth noting the processes of synaptic transmission, which may ultimately promote the triggering of a physiological answer in the individual [22–25]. This would explain the relationship between motivation and learning, due to the pleasure generated [26,27]. This way, pleasure is related to the exposure to happy stimuli, such as those caused by situations that include humor, which awaken in people an affective emotional bond due to the release of substances such as dopamine and serotonin, responsible for personal and emotional wellbeing [28,29]. Taking this into account, the educational benefits of using tools such as humor or surprise, which tend to elicit good memories and emotions among students, can (i) promote meaningful learning [30] and (ii) can improve student motivation, interest and performance [31,32].

Therefore, among the didactic tools or resources aimed at capturing students' attention to achieve the acquisition of knowledge, humorous resources play an important role, which, due to the emotional effects they generate in students, can facilitate learning [33]. In this sense, the importance of encouraging students through humorous resources has shown that emotions play a fundamental role in education, as well as in stress reduction [34], presenting a pedagogical perspective focused on psychological and neuroscientific aspects, such as learning processes, empathy and social development [35]. In addition, the use of humorous resources in the classroom has proven to be effective in improving classroom dynamics, because it qualitatively increases student participation and fosters their interest, thus developing their critical thinking [36,37].

The emergence of unconventional teaching resources such as memes is a direct consequence of the need for professors to motivate students through humorous resources to promote emotional development and improve the teaching–learning process [38,39]. A modern definition of a meme is that it is a piece of culture, typically a joke, which gains influence through online transmission [40,41]. In this sense, memes are a didactic resource of a humorous nature that has made its way into education in general, regardless of the educational stage, and has been used intensively as a learning tool in various disciplines, in order to facilitate the socialization of students, increase their motivation and participation, and thus obtain better learning results [42–44].

Given this scenario of virtualization of education and the progressive application of innovative humorous didactic resources, such as memes, to promote the teaching–learning process, the objective of this research is to analyze the perception of university professors about the employability and formative effectiveness of the use of humorous resources,

especially memes, in VLEs in higher education. Specifically, the study aims to quantitatively analyze the perception of university professors about the use of humorous resources in virtual learning environments and the usability of memes in this type of environment, and to identify gaps in these perceptions according to the professors' area of knowledge, gender, age or teaching experience. Specifically, it will be shown that professors give high ratings to humorous resources and memes in virtual learning environments and, in addition, that the area of knowledge is a discriminative variable of these perceptions that induces a gap in the answers of the participants of the study.

## 2. Materials and Methods

### 2.1. Participants

Participants were selected by a non-probabilistic convenience sampling process. The study involved 401 university professors from all areas of knowledge (Arts and Humanities, Sciences, Heath Sciences, Social and Legal Sciences and Engineering and Architecture) and from 19 different Latin American countries (Figure 1) (Argentina, Bolivia, Brazil, Chile, Colombia, Costa Rica, Cuba, Ecuador, El Salvador, Guatemala, Honduras, Mexico, Nicaragua, Panama, Paraguay, Peru, Dominican Republic, Uruguay and Venezuela), who attended a lecture on the use of humor strategies in higher education classes. After the presentation, a survey was sent to them as a Google Forms™ form, which they answered freely, voluntarily and anonymously. Therefore, all of them faced the survey with a homogeneous knowledge about the issue. All the attendees to the presentation answered the survey, and all the answers were qualified as valid. In the survey that was passed to the participants, they provided a series of data, in addition to their country of origin, which served to narrow down their sociological and academic profile, specifically: (i) gender, (ii) age, (iii) area of knowledge, and (iv) years of university teaching experience.

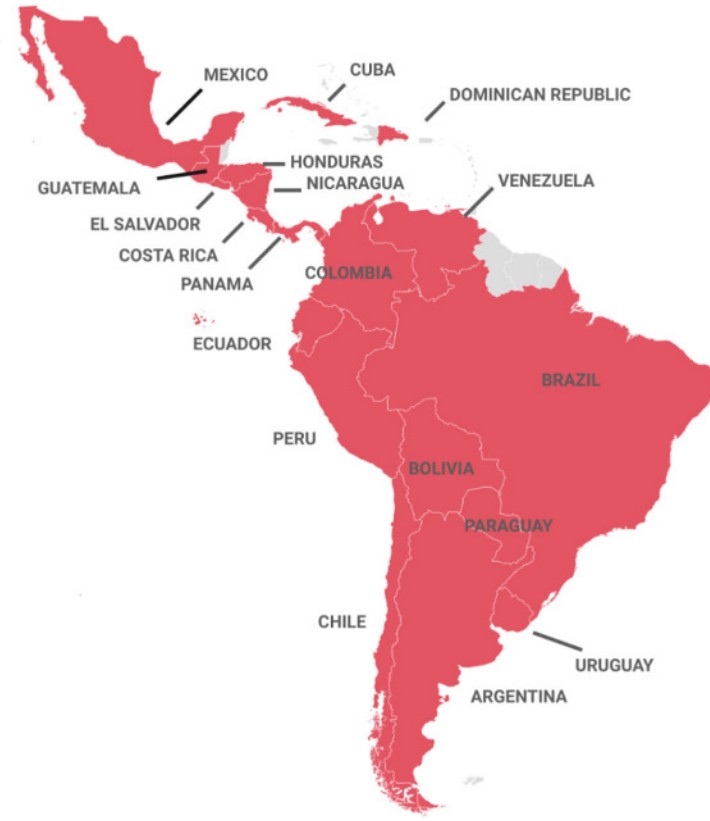

**Figure 1.** Latin American countries participating in the research.

### 2.2. Objectives, Variables and Hypothesis

The main objective of the study was to analyze the perception that Latin American university professors (Figure 2) have about the employability and formative effectiveness of the use of humorous resources (especially memes) in VLEs in higher education. Specifically, the following objectives are pursued: (i) to study the overall mean valuation of university professors about the use of humorous resources in classes developed in virtual environments and the dispersion of their answers, (ii) to explore the participants' opinion about the employability of memes as a didactic resource and their influence on the development of virtual classes and the academic performance of students, and (iii) to find gaps in the above valuations by reason of the participants' area of knowledge.

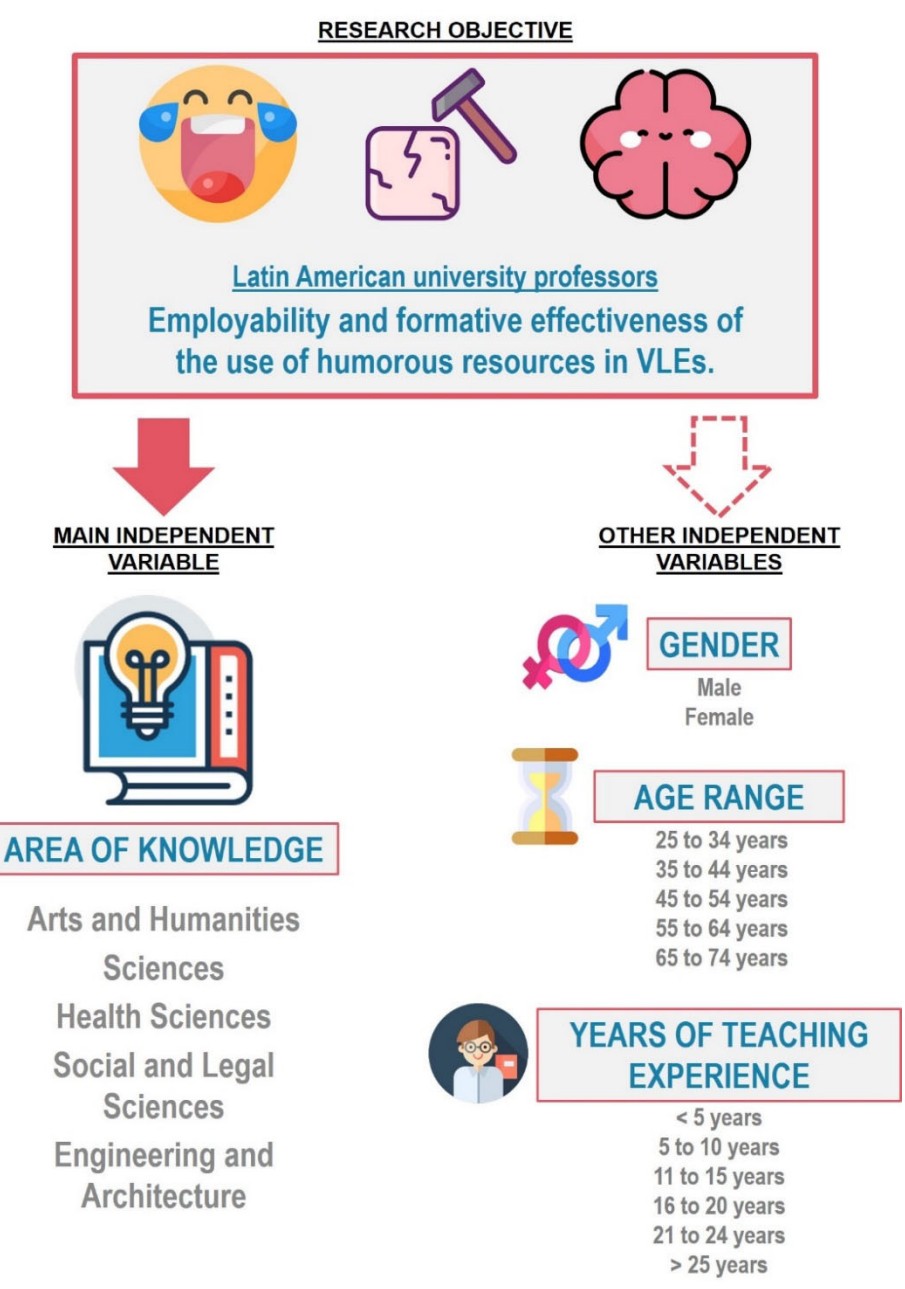

**Figure 2.** Research objective and independent variables.

The following explanatory variables were considered in the analysis, which delimit the sociological and academic profile of the participants: (i) area of knowledge, (ii) gender, (iii) age, and (iv) years of university teaching experience (Figure 2). The main discriminant

variable of the study was the area of knowledge, which is nominal and polytomous and can reach the following values: Arts and Humanities, Sciences, Health Sciences, Social and Legal Sciences, and Engineering and Architecture. The gender variable is dichotomous, ages have been grouped into five ranges of 10 years each, from 25 to 74 years old, and teaching experience has been grouped into five ranges of 5 years each, from 0 to 25 years of experience, together with a last range for those with more than 25 years of experience. Therefore, the variables age and teaching experience are polytomous nominals and reach five and six different values, respectively.

Likewise, the answer variables indicated in Table 1 were measured and grouped into the two large theoretical families indicated, which differentiate the variables about the didactic use of humor in general in virtual environments from those that refer specifically to the use of memes in the classroom. All the answer variables are quantitative, ordinal, and discrete and were valued on a scale of 1 to 5. The following hypotheses will be verified throughout the study:

**Hypotheses 1 (H1).** *Professors give high mean valuations to the use of humorous resources and memes in virtual higher education environments.*

**Hypotheses 2 (H2).** *There are gaps by area of knowledge in the valuation of humor and the use of memes in virtual environments by university professors.*

**Table 1.** Dependent variables.

| Type of Variables | Dependent Variable |
|---|---|
| Assessment of humor as a didactic resource in VLEs | • Connection with students' interests |
| | • Use of surprise as a didactic resource |
| | • Influence of humor in defining the professor's role in the classroom |
| | • Benefits of humor in student participation |
| | • Increasing student motivation |
| Use of memes in the virtual classroom | • Influence on connection with students |
| | • Influence on students' attention spans |
| | • Benefit on expected academic performance |
| | • Benefit on the progress of classes |
| | • Degree of compatibility of memes with the academic rigor of classes |
| | • Ease of use of memes in the classroom |

### 2.3. Instrument

To carry out the study, participants were given a survey with 11 Likert-type questions ranging from 1 to 5, where 1 corresponds to the lowest level of valuation or agreement, and 5 is the highest level of valuation or agreement. The first five questions assess the different aspects of the first set of answer variables, shown in Table 1, on the use of humor in university classrooms, and the last six assess the same, with the set of answer variables measuring the use of memes in the classroom. Each question therefore refers to each of the

answer variables in the sets of variables described above. The survey also measured the values of the sociodemographic variables shown in Figure 2 (area of knowledge, gender, age, and teaching experience). The data obtained have been treated statistically, descriptively and inferentially through ANOVA and MANOVA tests for comparison of mean data. Specifically, the answers were analyzed, differentiating by area of knowledge as the main discriminating variable and, secondarily, by the rest of the independent variables (gender, age and teaching experience).

*2.4. Procedure*

In this work, quantitative empirical research was carried out based on the data extracted from the survey that have been defined as an instrument. The survey was sent to the participants by e-mail through Google Forms[TM], and the statistical analysis was carried out using R statistical software. After obtaining the answers of the participants to the survey and checking their validity, an Exploratory Factor Analysis (EFA) was performed to identify the latent factors that explain the results and to define subscales within the instrument. Next, the validation of the instrument was concluded based on the parameters of the Confirmatory Factor Analysis (CFA), the computation of Cronbach's alpha parameters and the Pearson correlation measures of the different subscales with each other and with the global instrument.

Once the survey had been validated, descriptive statistics (mean and standard deviation) were extracted globally and were differentiated by the participants' area of knowledge, in order to perform a descriptive analysis of the results. The ANOVA test was used to identify gaps in the participants' evaluations by area of knowledge, and the multifactor ANOVA test (MANOVA) was used to find differences in these same evaluations when the area of knowledge variable was crossed with the rest of the explanatory variables. All mean comparison tests were performed with Welch's correction, without assuming homoscedasticity. Using Levene's and Bartlett's tests, it was found that there is homoscedasticity in the distribution of the answers when differentiated by the different independent variables. In all tests, 0.05 was taken as the significance level.

## 3. Results

### 3.1. Data Analysis

The sample of participants was distributed by areas of knowledge according to the percentages indicated in Table 2. This distribution is approximately homogeneous between the humanistic–social areas (Arts and Humanities and Social and Legal Sciences) and the scientific–technical areas (Sciences, Health Sciences and Engineering and Architecture). Conversely, within each group of areas, Social Sciences and Engineering professors were more frequent, respectively. In all areas, females and participants of central age and experience ranges were more frequent.

Regarding the instrument used, an EFA was performed on the answers with rotated factor loadings (Varimax rotation). The results of this analysis, which is aimed at identifying the latent subscales in the survey from the computation of their factor weights, are shown in Table 3. As a result, three latent factors were found to explain the answers obtained (chi-square = 25.29; df = 17; *p* value = 0.0884). In Table 3, the subscales corresponding to each factor are defined, and it indicates the highest factor weights for each item. As can be seen, all of them are greater than 0.5. This model explains 59.4% of the total variance, as shown in Table 4. Moreover, the Cronbach's alphas of the first two subscales (Table 3) are optimal and that of the third reaches an acceptable value, such that the survey has an adequate level of internal consistency.

**Table 2.** Distribution of participants by areas of knowledge, gender, age and teaching experience, in percentages (%).

|  |  | Humanities (%) | Science (%) | Health (%) | Soc. Sci. (%) | Engineering (%) |
|---|---|---|---|---|---|---|
| Global |  | 21.45 | 12.22 | 11.97 | 31.67 | 22.69 |
| Gender | Male | 36.0 | 30.6 | 35.4 | 33.9 | 44.0 |
|  | Female | 64.0 | 69.4 | 64.6 | 66.1 | 56.0 |
| Age | 25 to 34 years old | 15.1 | 16.3 | 14.6 | 15.7 | 5.5 |
|  | 35 to 44 years old | 26.7 | 30.6 | 27.1 | 24.4 | 34.1 |
|  | 45 to 54 years old | 34.9 | 36.7 | 29.2 | 30.7 | 33.0 |
|  | 55 to 64 years old | 18.6 | 16.3 | 18.8 | 25.2 | 18.7 |
|  | 65 to 74 years old | 4.7 | 0.0 | 10.4 | 3.9 | 8.8 |
| Experience | <5 years | 17.4 | 22.4 | 18.8 | 21.3 | 13.2 |
|  | 5 to 10 years | 19.8 | 20.4 | 20.8 | 25.2 | 30.8 |
|  | 11 to 15 years | 19.8 | 16.3 | 16.7 | 20.5 | 17.6 |
|  | 16 to 20 years | 10.5 | 14.3 | 25.0 | 11.0 | 14.3 |
|  | 21 to 24 years | 15.1 | 10.2 | 6.2 | 5.5 | 9.9 |
|  | >25 years | 17.4 | 16.3 | 12.5 | 16.5 | 14.3 |

The CFA confirms the model obtained from the EFA (chi-square = 90.7276; df = 41; $p$ value = 0.0000). The incremental fit indices obtained are optimal (AGFI = 0.9009; NFI = 0.9308; TLI = 0.9469; CFI = 0.9604; IFI = 0.9608), and the absolute fit indices also report that the model is adequate (GFI = 0.9385; RMSEA = 0.0699; AIC = 140.7276; chi-square/df = 2.2129).

As for the psychometric validation of the survey, the Pearson correlations (Table 5) show that there is a weak level of correlation between the different subscales defined by the model that emerges from the factor analysis. The highest correlation is established between the subscales of the use of humor in the virtual environment and the didactic benefits of memes, and yet this indicates a weak correlation. However, the correlations of each subscale with the overall scale defined by the complete survey are moderate to high. All correlations are statistically significant.

**Table 3.** EFA results, definition of subscales and Cronbach's alpha parameters.

| Item | Factor 1 Humor | Factor 2 Benefits of Memes | Factor 3 Memes Usability | Cronbach's Alphas |
|---|---|---|---|---|
| Connection with students' interests | 0.5920 |  |  |  |
| Use of surprise as a didactic resource | 0.6650 |  |  |  |
| Allows the professor's role to be adequately defined | 0.6190 |  |  | 0.8147 |
| Benefits in student participation | 0.7290 |  |  |  |
| Increasing student motivation | 0.7130 |  |  |  |
| Influence of memes on connection with students |  | 0.8790 |  |  |
| Memes promote attention |  | 0.8080 |  |  |
| Memes promote learning |  | 0.6320 |  | 0.9005 |
| Memes help the development of classes |  | 0.8510 |  |  |
| Memes are compatible with academic rigor |  |  | 0.7010 |  |
| Memes can be easily included in the classroom |  |  | 0.5810 | 0.6854 |

**Table 4.** Proportion and cumulative proportion of explained variance of the principal component analysis.

|  | Humor | Benefits of Memes | Memes Usability |
|---|---|---|---|
| Proportion of variance | 0.100 | 0.259 | 0.234 |
| Cumulative of variance | 0.100 | 0.359 | 0.594 |

**Table 5.** Pearson correlation coefficients of the different subscales among themselves and with respect to the global scale.

|  | Humor | Benefits of Memes | Memes Usability | Global |
|---|---|---|---|---|
| Humor | 1 | 0.3793 | 0.0575 | 0.5323 |
| Benefits of memes |  | 1 | 0.1339 | 0.6682 |
| Memes usability |  |  | 1 | 0.7657 |
| Global |  |  |  | 1 |

*3.2. Subscale of Evaluation of Humor as a Didactic Resource in VLEs*

The participants rate the use of humor in virtual classrooms highly (mean 4.53 out of 5) with a low dispersion in the answers (standard deviation = 0.70). By areas of knowledge, the humanistic–social fields give higher mean valuations (4.63 out of 5 in Humanities and 4.55 out of 5 in Social Sciences) than the scientific–technical fields (4.35 out of 5 in Sciences, 4.53 out of 5 in Health Sciences and 4.49 out of 5 in Engineering and Architecture). The ANOVA test shows that these differences are significant (F = 6.8520, $p$ = 0.0000).

Among males, Science professors rate humor the lowest and Social Sciences the highest, while Health Sciences professors surpass Humanities professors in their mean valuation of humor (Table 6). Conversely, among females, it is the Engineering professors who give the lowest score to the use of humor and those in the Humanities the highest. Contrary to what happens with males, female professors in the Sciences outperform those in the Social Sciences in their evaluation of humor. The MANOVA test confirms that these gaps by area of knowledge within each gender are statistically significant (F = 10.6408; $p$ = 0.0000).

Differentiating by age (Table 7), among the youngest participants, the professors of Health Sciences surpass those of the humanistic–social areas and those of Sciences slightly surpass those of Social Sciences in their valuation of humor. Conversely, among participants aged 35 to 44 years, only those from the Health Sciences outperformed those from the Social Sciences and, in any case, those from the Humanities outperformed professors from the other areas. Among those over 45 years of age, the highest scores are again reached by Humanities professors, except among those over 65 years of age, among whom the highest scores are given by Social Sciences professors, followed by Engineering professors. Except among professors of extreme ages (the youngest and the oldest), the area in which humor is most highly rated is the Humanities. All these differences are statistically significant (F = 4.5770; $p$ = 0.0000).

When differentiated by years of teaching experience, the results are slightly different from those obtained by age, although the differences identified in the ratings differentiated by years of experience are still statistically significant (F = 6.3843; $p$ = 0.0000). Table 8 shows that the highest ratings of humor are obtained in the humanistic–social areas in all ranges of experience, except among those with less than 10 years of experience or between 21 and 25 years (professors of Health Sciences have the highest ratings).

**Table 6.** Mean values of the scale on the didactic use of humor in VLEs, differentiated by areas of knowledge and gender (ratings out of 5).

|  | Males | Females |
|---|---|---|
| Humanities | 4.55 | 4.67 |
| Science | 3.95 | 4.53 |
| Health | 4.59 | 4.50 |
| Soc. Sci. | 4.61 | 4.52 |
| Engineering | 4.52 | 4.46 |

**Table 7.** Mean values of the scale on the didactic use of humor in VLEs, differentiated by areas of knowledge and age ranges (ratings out of 5).

|  | 25 to 34 Years Old | 35 to 44 Years Old | 45 to 54 Years Old | 55 to 64 Years Old | $\geq$65 Years Old |
|---|---|---|---|---|---|
| Humanities | 4.57 | 4.73 | 4.60 | 4.64 | 4.45 |
| Science | 4.53 | 4.24 | 4.48 | 4.10 | - |
| Health | 4.83 | 4.69 | 4.53 | 4.58 | 3.64 |
| Soc. Sci. | 4.51 | 4.55 | 4.59 | 4.51 | 4.72 |
| Engineering | 4.48 | 4.52 | 4.41 | 4.52 | 4.60 |

**Table 8.** Mean values of the scale on the use of humor, differentiated by areas of knowledge and years of teaching experience (ratings out of 5).

|  | $\leq$5 Years | 6 to 10 Years | 11 to 15 Years | 16 to 20 Years | 21 to 25 Years | >25 Years |
|---|---|---|---|---|---|---|
| Humanities | 4.64 | 4.69 | 4.75 | 4.60 | 4.40 | 4.63 |
| Science | 4.45 | 3.80 | 4.65 | 4.66 | 3.88 | 4.63 |
| Health | 4.76 | 4.82 | 4.50 | 4.48 | 4.60 | 3.83 |
| Soc. Sci. | 4.52 | 4.59 | 4.48 | 4.50 | 4.57 | 4.65 |
| Engineering | 4.47 | 4.61 | 4.41 | 4.37 | 4.44 | 4.48 |

Among professors with less than 10 years of experience, those with the lowest mean value for humor are those in the sciences. After 11 years, this position is occupied by Engineering professors, Science professors between 21 and 25 years and Health Sciences after 25 years of experience. The valuation made by Health Sciences professors with less experience is the highest of all areas, while it is the lowest of all areas among those with more than 25 years of experience. The mean value of the use of humor among Health Sciences professors is approximately decreasing as teaching experience increases, a phenomenon that does not occur with such clarity in any other area.

*3.3. Benefits of Memes Subscale*

The educational benefits of the use of memes in university virtual classrooms reach a high mean valuation (4.24 out of 5), although slightly lower than the mean valuation of the use of humor in general. The standard deviation is also small (0.95 out of 5, which is less than a quarter of the mean) but somewhat higher. The ANOVA test identifies significant differences by area of knowledge (F = 5.5430; *p* = 0.0002). Science professors give the lowest mean valuation (3.96 out of 5), followed by Health Sciences (4.20 out of 5) and Engineering (4.26 out of 5). As in the previous scale, Humanities (4.27 out of 5) and Social Sciences (4.55 out of 5) professors give the highest mean valuation. In general, the mean valuations of the benefits of memes are lower than those of the use of humorous resources in all areas of knowledge.

The MANOVA test found no statistically significant differences by gender (F = 1.1205; *p* = 0.3451). However, there are significant differences between areas of knowledge when differentiated by age ranges (F = 4.5574; *p* = 0.0000), whose mean values are shown in Table 9. Among those under 45 years old, the highest valuation of the benefits of memes is given by Humanities professors. However, above that age, the highest ratings are reached among professors of Social Sciences and Health Sciences (between 45 and 54 years old) or Engineering (55 to 64 years old). In the case of those over 65 years old, the lowest ratings correspond to Humanities and Health Sciences and the highest to Social Sciences and Engineering (there are no participants from Sciences in this age group).

Table 9. Mean values of the scale on the didactic benefits of memes in virtual environments, differentiated by areas of knowledge and age ranges (ratings out of 5).

| | 25 to 34 Years Old | 35 to 44 Years Old | 45 to 54 Years Old | 55 to 64 Years Old | $\geq$65 Years Old |
|---|---|---|---|---|---|
| Humanities | 4.81 | 4.48 | 4.06 | 4.06 | 3.75 |
| Science | 4.00 | 3.83 | 3.96 | 4.16 | - |
| Health | 4.68 | 4.33 | 4.25 | 4.28 | 2.95 |
| Soc. Sci. | 4.50 | 4.30 | 4.33 | 4.18 | 4.70 |
| Engineering | 4.35 | 4.38 | 4.06 | 4.37 | 4.31 |

There are also statistically significant differences when differentiated by years of teaching experience (F = 6.7325; $p$ = 0.0000). The most significant results in this regard (Table 10) are the relatively low scores of Science professors (except for professors with more than 25 years of experience), the low ratings of Humanities professors with 11 to 15 or more than 20 years of experience, and the low ratings of Health Sciences and Engineering professors with more than 25 years of experience.

Table 10. Mean values of the scale on the didactic benefits of memes in virtual environments, differentiated by areas of knowledge and teaching experience (ratings out of 5).

| | $\leq$5 Years | 6 to 10 Years | 11 to 15 Years | 16 to 20 Years | 21 to 25 Years | >25 Years |
|---|---|---|---|---|---|---|
| Humanities | 4.67 | 4.43 | 3.99 | 4.67 | 3.94 | 4.07 |
| Science | 4.20 | 3.05 | 4.09 | 4.21 | 3.75 | 4.53 |
| Health | 4.69 | 4.60 | 4.28 | 4.00 | 4.17 | 3.13 |
| Soc. Sci. | 4.52 | 4.38 | 4.38 | 4.14 | 4.00 | 4.15 |
| Engineering | 4.69 | 4.40 | 4.05 | 4.27 | 4.25 | 3.85 |

*3.4. Memes Usability Subscale*

Despite the high valuation given by the participants to the didactic benefits of the use of memes, the usability of memes in the virtual classroom is evaluated in an intermediate way by the participants (mean 3.74 out of 5). However, in this subscale, there is a higher dispersion of answers than in the other subscales (standard deviation 1.43 out of 5), which shows that the professors' opinion of the usability of memes is not as solidly formed as that of their didactic benefits. In this sense, the ANOVA test does not identify significant gaps by areas of knowledge (F = 1.5770; $p$ = 0.1780), which indicates that the above results are homogeneous among participants from different areas. Likewise, the MANOVA test also does not identify gaps, within the different areas of knowledge, by gender (F = 0.2348; $p$ = 0.9188), age (F = 1.1474; $p$ = 0.3091) or teaching experience (F = 1.5586; $p$ = 0.05636).

**4. Discussion**

In the present work, an instrument has been designed to measure the perception of a group of university professors about the use of humorous resources, especially memes, in virtual learning environments. The instrument has been validated through different ways, including factor analysis, internal consistency and psychometric dimensions (Tables 3–5). As a result of this validation process, three fundamental dimensions present in the analyzed perception have been distinguished: (i) didactic use of humor, (ii) perceived benefits in the use of memes in the classroom, and (iii) usability of memes in higher education (Table 3). By analyzing the results, it has been found that the valuation given by university professors to the use of humor resources in VLEs in higher education is positive (Section 3.2). This valuation is also elevated when analyzing the benefits of the use of memes in VLEs (Section 3.3). Therefore, the first two specific objectives of the paper have been met, and hypothesis H1 can be considered as verified.

Some theoretical studies point out the reasons that may be behind the above results: (i) the ability of humor to capture the attention and interest of students, while focusing

that attention on the contents developed [45] or (ii) the instructive function of humor, in the sense of generating relaxed learning contexts in which the student feels comfortable [11,31,32,46–48]. This high valuation of humor as a didactic resource agrees with the results of several previous studies [49,50], which analyze the perspective of university professors on the use of humor in the classroom and report high general valuations in this regard.

In the specialized literature on the use of humor in higher education classrooms, there is an abundance of research based on the description of specific classroom experiences and the analysis of learning outcomes derived from the use of humor and student perception [51–54], both in traditional and virtual environments [55]. All of them coincide with highlighting the interest of humorous resources for the understanding of complex concepts and for increasing student motivation, especially in VLEs.

Studies that analyze the professors' perspective on the use of humorous resources are less numerous, and none of them are focused on VLEs. Some authors highlight the value that professors place on humor as a factor that decreases stress toward learning complex concepts [56]. Therefore, the results obtained in the present research confirm that the professors' perspective on the use of humor in VLEs is in line with the perspective they express in traditional environments.

It has also been shown that there are significant gender differences in the opinions on the use of humor as a didactic resource. In this sense, there is consonance with the results of previous research, such as [57], where it is shown that female professors give, in general, a higher valuation of humor in the classroom. This work shows that, when focusing attention on VLEs, it is not enough to consider the gender variable to achieve a complete explanation of the differences between opinions, but it is necessary to combine gender with the area of knowledge (Table 6). Thus, in the areas of Sciences and Humanities, female professors offer higher ratings than males, while in the rest of the areas, the highest ratings correspond to males. Likewise, it was found that the variables gender and area of knowledge are not the only variables that discriminate against professors' opinions, but that age and teaching experience also discriminate against them (Tables 7 and 8), which is a novelty with respect to previous research. In any case, professors' opinions by age or time of experience change significantly with the area of knowledge, which makes the latter variable the most discriminative. While in the areas of Humanities, Health Sciences and Engineering, the assessment is approximately decreasing with age, the opposite effect occurs in the areas of Social Sciences and Sciences (Table 7). The results by teaching experience are not absolutely analogous, but it is also perceived that the areas in which the most experienced professors are the ones who outperform the less experienced ones are those of Sciences and Social Sciences (Table 8). This is probably strongly linked to the fact that the learning context is virtual, but there are no other analogous studies with which to compare results and it would be necessary to go deeper in this sense to corroborate this hypothesis.

Regarding the use of memes in virtual environments, the results obtained confirm that the professors' assessment is high in general (Section 3.3). In this sense, the results are in line with those of other similar studies conducted in traditional learning environments [39,53,58]. This study also shows that the area of knowledge is once again a discriminating variable in the perception of professors in this regard, with Humanities and Social Sciences professors giving the highest value to memes in virtual environments. By age and teaching experience, there is a certain drop in the ratings as age and experience increase, except in the area of Sciences. Again, a specific analysis would be needed to identify the reasons for this phenomenon. In view of the above, it can be concluded that the third specific objective of this research has been satisfied.

Although the valuation of memes in virtual environments is high, the participants give an intermediate opinion about their usability in virtual environments in higher education (Section 3.4). For this reason, it can be concluded that hypothesis H2 of the present research has been partially verified. In some previous studies [49], it is pointed out, regarding the use of humor as a didactic resource, that although its valuation is high, professors

do not use it spontaneously in the classroom. This could explain why, in this study, the perception of the usability of memes in virtual environments is lower than the assessment of their effectiveness as a teaching resource. In this same sense, some research indicates that, before using memes, the professor should make an analysis exercise about their own ability to design these resources, to illustrate difficult concepts through them and to propose activities to students based on them [57]. Other authors emphasize the difficulty of satisfying certain learning objectives with humorous resources, especially memes [59], or the ethical lines that could be crossed if this type of resource is misused (e.g., the risk of offending people or social groups) [60]. In addition, there is an obvious risk of turning humorous resources, such as memes, into mere objects for students' amusement, without effectively aiding learning [61]. This danger is especially present in VLEs, since they reduce the capacity for interaction between the professor and the students. All these reasons may explain the intermediate assessment that professors give to the usability of memes in virtual environments of higher education.

## 5. Conclusions

The use of humorous resources in university classrooms is a topic of interest in current research on innovative didactic trends in higher education. Proof of this is the abundance of research papers that have been published on the subject in recent years. However, the professors' perspective on these didactic resources and their application in virtual learning environments (VLEs) are not usually the most common approaches. This research explored the opinion of 401 university professors about the use of humor resources, especially memes, in virtual classrooms of higher education.

The results obtained confirm that the ratings of these resources are high, similar to those reported by other research studies contextualized in traditional learning environments. In addition, the area of the professor's knowledge has been identified as a discriminating variable of these perceptions.

The fact that the usability of memes is less valued, in general, than their didactic effectiveness suggests the need for specific training actions on the use of humorous resources, especially memes, for university professors. These training actions should be aimed at providing digital tools for the design of these resources and criteria for learning how to design them, in the sense of projecting the concepts of the contents of the subjects in these resources. It would be convenient to carry out future research work to identify the reasons for the differences between the opinions of professors of different areas of knowledge. This would make it possible to focus the suggested training actions specifically on the different areas of knowledge. Other lines of future research consist of considering other sociodemographic variables, such as country of origin, in the quantitative analysis.

**Author Contributions:** Conceptualization, Á.A.-S. and D.V.-R.; methodology, Á.A.-S. and D.V.-R.; validation, Á.A.-S., P.F.-A. and D.V.-R.; formal analysis, Á.A.-S.; data curation, Á.A.-S., M.N.-S., P.F.-A. and D.V.-R.; writing—original draft preparation, Á.A.-S., M.N.-S., P.F.-A. and D.V.-R.; writing—review and editing, Á.A.-S., M.N.-S., P.F.-A. and D.V.-R.; supervision, Á.A.-S., M.N.-S., P.F.-A. and D.V.-R. All authors have read and agreed to the published version of the manuscript.

**Funding:** This research received no external funding.

**Institutional Review Board Statement:** The protocol was approved by the Ethics Committee of the Project "Influence of COVID-19 on teaching: digitization of laboratory practices at UCAV" (1 September 2021).

**Informed Consent Statement:** All participants were informed about the anonymous nature of their participation, why the research is being conducted, how their data will be used and that under no circumstances would their data be used to identify them.

**Data Availability Statement:** The data are not publicly available because they are part of a larger project involving more researchers. If you have any questions, please ask the contact author.

**Conflicts of Interest:** The authors declare no conflict of interest.

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
