# Peer review of "Usability of Memes and Humorous Resources in Virtual Learning Environments"

_education, doi:10.3390/educsci12030208_

Round 1

Reviewer 1 Report

I appreciate the author's contribution to the rapidly developing field of virtual learning, but the manuscript must be formatted according to the journal guidelines prior to publication.

Point 1. Name and version of any software used should be included into “Materials and Methods” section.

Point 2. For non-interventional studies (e.g. surveys, questionnaires, social media research), all participants must be fully informed if the anonymity is assured, why the research is being conducted, how their data will be used and if there are any risks associated. As with all research involving humans, ethical approval from an appropriate ethics committee must be obtained prior to conducting the study. If ethical approval is not required, authors must either provide an exemption from the ethics committee or are encouraged to cite the local or national legislation that indicates ethics approval is not required for this type of study. Where a study has been granted exemption, the name of the ethics committee which provided this should be stated in Section ‘Institutional Review Board Statement’ with a full explanation regarding why ethical approval was not required.

Yours faithfully,

Reviewer

Author Response

Please, find the response in the attached document

Reviewer 2 Report

1. The literature review is well structured.

2. English language is adequate.

3. Statistical analysis is performed, but we recommend introducing an additional variable (country name -id) to analyze if the are any statistical differences between countries.

Author Response

(The authors gave the same response as above.)

Reviewer 3 Report

It is an interesting research and worked methodologically.
It would be interesting in the introduction to include hypotheses,
as well as secondary objectives that support the main objective
and the methodology. The objectives that are in section 2.2.,
must go in the introduction, as well as the hypothesis.
In the participants section, a table with the participants classified
by country, gender, specialty, etc. would be interesting. Table 1 should
explain the instrument, which should be more detailed, including
sociodemographic data from the survey, as well as the type of
methodology used. From Table 6 and section 3.2. they are data
differentiated by gender, areas of knowledge, age, years of experience,
which would be interesting if they would be collected in the previous
data of the 11 items of the questionnaire, in Table 3. From Table 6,
it is unknown if these data they were removed from the questionnaire,
if they were contemplated in the questionnaire, because they are not
mentioned in the description of the instrument. And the discussion
focuses, above all, on the results from Table 6, there is a lack of
further discussion on Table 3, which are the elements of the
questionnaire.

Author Response

(The authors gave the same response as above.)
